# Geographic Setting and Groundwater Table Control Carbon Emission from Indonesian Peatland: A Meta-Analysis

**Nisa Novita** [1,*] **, Nurul Silva Lestari** [2] **, Mega Lugina** [2] **, Tatang Tiryana** [3] **, Imam Basuki** [4] **and Joni Jupesta** [5]

1 Yayasan Konservasi Alam Nusantara, Graha Iskandarsyah, Jl. Iskandarsyah Raya No. 66 C, Jakarta Selatan, Jakarta 12160, Indonesia

2 Center for Research and Development of Socio-Economic Policy and Climate Change, Ministry of Environment and Forestry, Jalan Gunung Batu No. 5, West Java, Bogor 16118, Indonesia; nurulsilva@gmail.com (N.S.L.); megalugina76@gmail.com (M.L.)

3 Department of Forest Management, Faculty of Forestry and Environment, Kampus IPB Dramaga, IPB University, West Java, Bogor 16680, Indonesia; tangtir@apps.ipb.ac.id

4 Yayasan Wineco Indonesia Lestari-Winrock International, Menara Mandiri Tower 2, 17th Floor, Jl. Jend. Sudirman, Kav. 54-55, Jakarta Selatan, DKI Jakarta 12190, Indonesia; imambasuki1974@gmail.com

5 Research Institute of Innovative Technology for the Earth, 9-2 Kizugawadai, Kizugawa 619-0292, Japan; jjupesta@yahoo.com

* Correspondence: nisa.novita@tnc.org; Tel.: +622-1727-9204

**Abstract:** Peat restoration is a key climate mitigation action for achieving Indonesia's Nationally Determined Contribution (NDC) emission reduction target. The level of carbon reduction resulting from peat restoration is uncertain, owing in part to diverse methodologies and land covers. In this study, a meta-analysis was conducted to assess the impact of rewetting on reduction of total $CO_2$ in soil and heterotrophic emissions at the country level. The tier 2 emission factor associated with the land cover category in Indonesia was also calculated. The analysis included a total of 32 studies with 112 observations (data points) for total $CO_2$ emissions and 31 observations for heterotrophic emissions in Indonesia. The results show that the land cover category is not a significant predictor of heterotrophic and total soil emissions, but the highest observed soil emissions were found in the plantation forest. Using the random-effects model, our results suggest that an increase in the water table depth of 10 cm would result in an increase in total $CO_2$ emissions of 2.7 Mg $CO_2$ ha$^{-1}$ year$^{-1}$ and an increase in heterotrophic emissions of 2.3 Mg $CO_2$ ha$^{-1}$ year$^{-1}$. Our findings show that managing water table depth in degraded peatlands in various land cover types is important to achieve Indonesia's emission reduction target by 2030.

**Keywords:** $CO_2$; emissions factor (EF); Indonesia; meta-analysis; peatland; water table

## 1. Introduction

Protecting and restoring soil organic matter delivers many benefits for people and provides a comprehensive solution for climate change, in particular for tropical peatlands [1,2]. There is a growing international interest in soil carbon in international climate mitigation work, such as the "4 per 1000" Initiative in Paris in 2015 and recognition of soil organic carbon (SOC) sequestration in the United Nations Framework Convention on Climate Change (UNFCCC) process in 2017 in the COP 23 decision 4/CP.23. The SOC element with the highest potential for natural climate solutions (NCSs) in the tropics is peatland restoration, which stands at 200 GtCO$_2$eq year$^{-1}$ [3,4]. Specifically, NCS analysis has shown that restoring peatlands is one of the most promising strategies to achieve country emission reductions by 2030 [5], with potential emission reductions of 878 MtonCO$_2$eq year$^{-1}$ in Indonesia [4].

Peatlands are critical for climate change and the global carbon cycle. However, the function of peatlands will be switched from sink to source in this century [6]. Moreover, undrained tropical peatlands have a significant climate stabilizing effect because of the

rich carbon underneath the soil [7,8]. Tropical peatlands represent an important ecosystem in the global carbon budget, accounting for 10% of global peatlands and storing 50–350 GtC [9–11]. On the other hand, drained tropical peatlands, due to land use change through drainage and fires, have completely different effects, acting as significant contributors to global greenhouse gas emissions [12–15]. They are responsible for almost 25% of global carbon emissions from the land use sector [8]. Specifically in Indonesia, emissions from peat decomposition and fires contribute to 76% of the total agriculture and forestry annual emissions [16]. As the forestry sector is the main foundation of the NDC emission reduction target, and peatlands are the major contributor in the forestry sector, exploring the potential for Indonesia to extend and push the ambitious commitment over a longer period is necessary.

Indonesia already ratified the Paris Agreement in 2016 and submitted its Nationally Determined Contribution (NDC) in the same year. Based on Indonesia's NDC, either 1.4 million ha (Counter Measure 1 scenario) or 2.9 million ha (Counter Measure 2 scenario) of degraded peatlands will be restored within the period from 2014 to 2030. Indonesia is known as the second largest tropical peatland forest in the world, with 14.9–22 million ha of peatland [11,17,18]. Therefore, to achieve the emissions reduction target, Indonesia has pledged to restore two million ha of degraded peatland [19] and established the Peat Restoration Agency in 2015, which has now been extended to become the Peat and Mangrove Restoration Agency. Rewetting, as the main component in the restoration program, should be properly quantified with robust scientific evidence. Rewetting is not only useful to restore degraded peatlands, but also to protect remaining intact forests from fire risks [17,20,21]. Studies from tropical peatland types have demonstrated that increasing the water table through rewetting reduces $CO_2$ emissions and subsidence [22–25].

Hoyt et al. [26] have observed an effect of soil temperature and moisture on soil heterotrophic respiration. Furthermore, Cobb et al. [27] concluded that rainfall seasonality can affect the $CO_2$ emissions from tropical peatlands. Despite the importance of conserving and restoring peatlands in climate change mitigation, data on the relationship between soil GHG emissions and environmental variables in tropical peatlands is limited. Hooijer et al. [22] have provided several equations to estimate carbon emissions from water table fluctuation using eight sampling points in Riau, Sumatra. These equations were used as an approved Verified Carbon Standard (VCS) methodology for rewetting drained tropical peatlands (VM0027). In a recent review, Carlson et al. [25] used a linear regression model to determine the relation between water table depth and soil respiration. This model was built upon the IPCC's tier 1 emissions factor, which is based on limited sources (12 studies and 59 sites). In this study, we aimed to improve upon these previous studies by expanding the sampling numbers from various land use and cover types for peatlands and by testing the environmental variables at the country scale. This study had two objectives:

- To provide a Tier 2 emission factor estimate for peat decomposition using recent publications in Indonesia;
- To model the relationship between total and heterotrophic respirations with significant environmental predictors (i.e., land use, land cover class, geographical coordinate, water table depth, bulk density, and air temperature) in order to quantify $CO_2$ emission reductions from rewetting.

## 2. Methodology

### 2.1. Scope of the Study

This meta-analysis was based on 31 peer-reviewed journal papers and 1 project report. The research works were published between 2005 and 2019 from 112 study sites located in the Sumatra and Kalimantan islands of Indonesia, covering seven provinces: Aceh, North Sumatra, Riau, Jambi, West Kalimantan, South Kalimantan, and Central Kalimantan. Figure 1 depicts the geographical scope of this study. We classified the land use category based on the degree of degradation: cropping/shrubland, drained burnt, forest, and plantation. Land cover categories for observed peatlands in this study

were based on the categories from the Indonesian Ministry of Environment and Forestry (MoEF), land cover categories adapted from IPCC for wetlands supplement [28], and land management classes. A more detailed explanation about land cover categories can be found in Indonesia's first Forest Emissions Level Reference [29]. In addition, land cover categories—namely cropland and fallow, drained; cropland, drained, paddy rice; forestland and cleared forestland, drained; plantations, drained, oil palm; and plantations, drained, short rotations [28]—were also assessed in this study.

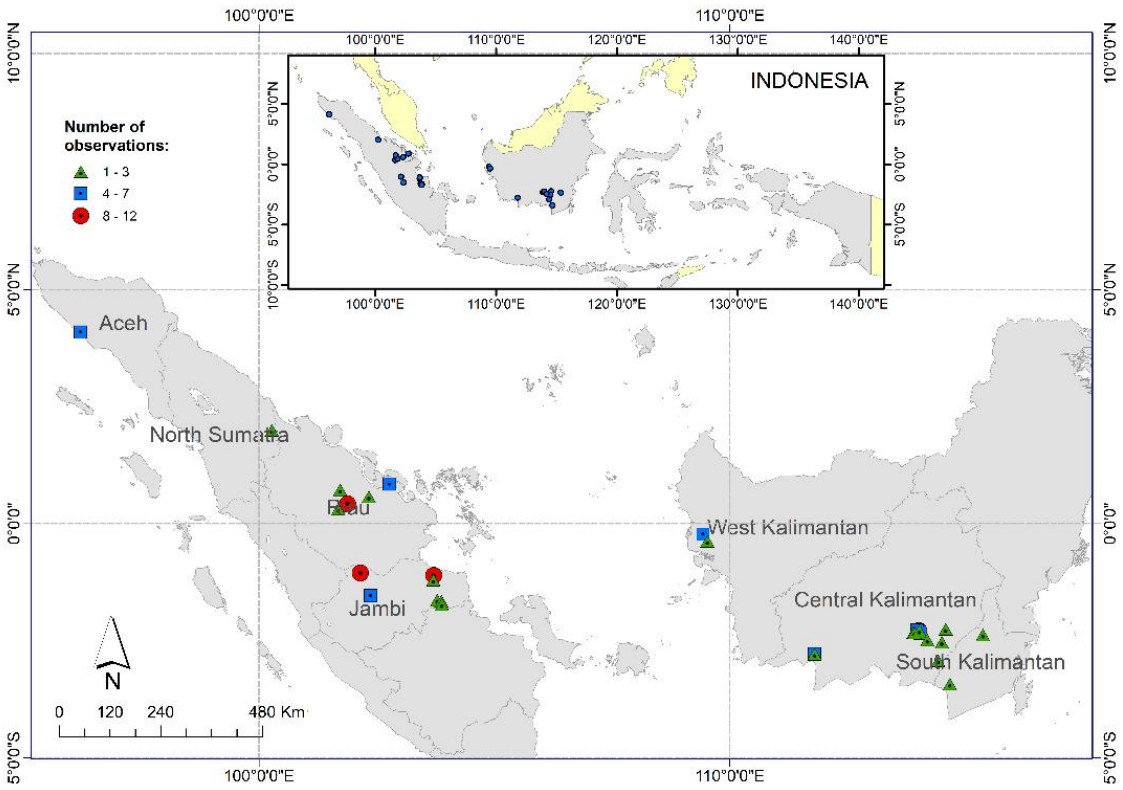

**Figure 1.** Geographical extent of the primary study areas used for the meta-analysis. Green diamonds, blue squares, and red circles denote the numbers of observations for 1–3 data observations, 4–7 data observations, and 8–12 data observations, respectively.

### 2.2. Total CO₂ and Heterotrophic Emissions Data Set

The dataset on total $CO_2$ and heterotrophic emissions was collected through a systematic review of publications of peatlands in Indonesia, as shown in Table 1 Additional data were also extracted from the publications to provide predictor variables (moderators) that might explain the heterogeneity of $CO_2$ emissions. Among others, the predictor variables used in this meta-analysis were geographical coordinates (latitude and longitude), land use class/land cover class, water table depth (cm), air temperature (°C), annual rainfall (mm year$^{-1}$), and bulk density (g cm$^{-3}$). Where necessary, the $CO_2$ emissions and predictor variables data were elicited by converting graphical data using the GetData Graph Digitizer (http://getdata-graph-digitizer.com (accessed on 23 February 2021)) and by accessing an online climate database (https://power.larc.nasa.gov/data-access-viewer (accessed on 24 February 2021)) when air temperature and annual rainfall data were absent in the publications. The details of the study titles and authors are provided in Table A1.

### 2.3. Emission Factor (EF) in Different Land Use and Land Covers Categories

The mean and SD of total $CO_2$ and heterotrophic emissions (Mg $CO_2$ ha$^{-1}$ year$^{-1}$) from each site were combined across the studies to derive numbers for the tier 2 level for each land use/land cover class category. The true value of total $CO_2$ and heterotrophic

emissions in each primary study remained unknown, but it was assumed to vary from one study area to another. The random-effects models with the restricted maximum-likelihood (REML) estimator and the Knapp and Hartung adjustment [30] were used to derive the mean total $CO_2$ and heterotrophic emissions (EF) at the tier 2 level with the "metafor" package of R version 3.6.3 [31,32]. The inter-study heterogeneity was assessed using the $I^2$ and Q statistics [30].

**Table 1.** List of the publications and number of observations for total $CO_2$ and heterotrophic emissions used in this meta-analysis [13,15,22,33–61].

| No. | Author (Year) | Number of Observations | |
|-----|---------------|-------------------------|----|
| | | Total $CO_2$ | Heterotrophic |
| 1 | Ali et al., (2006) [33] | 3 | |
| 2 | Astiani et al., (2018) [34] | 4 | |
| 3 | Batubara et al., (2019) [35] | 2 | |
| 4 | Comeau et al., (2016) [36] | 1 | |
| 5 | Dariah et al., (2014) [37] | 2 | 2 |
| 6 | Furukawa et al., (2005) [38] | 12 | |
| 7 | Hadi et al., (2005) [39] | 3 | |
| 8 | Handayani et al., (2009) [40] | 7 | 7 |
| 9 | Hergoualc'h et al., (2017) [41] | 3 | 3 |
| 10 | Hirano et al., (2007) [42] | 1 | |
| 11 | Hirano et al., (2009) [15] | 6 | |
| 12 | Hirano et al., (2014) [43] | 2 | |
| 13 | Hooijer et al., (2012) [22] | 2 | |
| 14 | Hooijer et al., (2014) [44] | 1 | |
| 15 | Husnain et al., (2014) [45] | 1 | 6 *) |
| 16 | Inubushi et al., (2003) [46] | 3 | |
| 17 | Ishikura et al., (2017) [47] | 11 | |
| 18 | Ishizuka et al., (2002) [48] | 8 | |
| 19 | Itoh et al., (2017) [49] | 3 | |
| 20 | Jamaludin et al., (2020) [50] | 3 | 3 |
| 21 | Jauhiainen et al., (2005) [51] | 1 | |
| 22 | Jauhiainen et al., (2008) [13] | 4 | |
| 23 | Jauhiainen et al., (2012) [52] | 8 | 8 |
| 24 | Khasanah and van Noordwijk (2019) [53] | 4 | |
| 25 | Marwanto and Agus (2014) [54] | 1 | |
| 26 | Marwanto et al., (2019) [55] | 1 | |
| 27 | Saragi-Sasmito et al., (2019) [56] | 1 | 1 |
| 28 | Sundari et al., (2012) [57] | 2 | |
| 29 | Swails et al., (2019) [58] | 6 | |
| 30 | Toma et al., (2011) [59] | 1 | |
| 31 | Wakhid et al., (2017) [23] | 1 | 1 |
| 32 | Watanabe et al., (2009) [60] | 4 | |
| | Total | 112 | 31 |

*) Husnain et al. (2014) provided 6 heterotrophic emissions data without their corresponding total $CO_2$ emissions for the same sites.

## 2.4. Model for Estimating Total $CO_2$ and Heterotrophic Emissions

The heterogeneity of $CO_2$ emissions can be influenced by various factors, including water table depth, land use types, microtopography, precipitation, temperature, and vegetation physiology [15,46]. Based on the data availability, this meta-analysis considered five predictor variables that might account for the heterogeneity of $CO_2$ emissions (*TE* or *HE*, Mg ha$^{-1}$ year$^{-1}$): water table depth (*W*, cm), air temperature (*T*, °C), annual rainfall (*R*, mm year$^{-1}$), geographical location (in terms of absolute latitude, *L*), and bulk density (*B*, g cm$^{-3}$). These predictor variables were used to estimate total $CO_2$ and heterotrophic emissions using the following linear mixed-effects model [31]:

$$Y_i = \beta_0 + \beta_1 X_{1i} + \beta_2 X_{2i} + \ldots + \beta_p B_{ki} + u_i \qquad (1)$$

where $Y_i$ is total or heterotrophic $CO_2$ emissions; $\beta_0$, $\beta_1$, ..., $\beta_p$ are regression parameters; $X_1$, $X_2$, ..., $X_k$ are predictor variables (i.e., *W*, *T*, *R*, *L*, or *B*); and $u_i$ indicates the random effects of the *i*-th study that were assumed to be normally distributed with mean μ and variance $\tau^2$. The "metafor" package was first used to generate a linear mixed-effects model using all predictor variables, which was then reduced into more simple models when some predictor variables were not found to be statistically significant. The maximum likelihood (ML) estimator and the Knap and Hartung adjustment methods were used to estimate the model parameters and their associated significant tests. The heterogeneity of total and heterotrophic $CO_2$ emissions accounted for by the predictor variables in each model was assessed using $R^2$ analog [30] and the comparison of model fits was based on the corrected Akaike Information Criterion (AICc) [31].

## 3. Results

### 3.1. Data Site

While all the primary studies provided total $CO_2$ emissions data, only eight primary studies (ca. 25%) provided heterotrophic emissions data (Table 1). Based on the spatial distribution shown in Figure 1, there were no primary studies conducted in Sulawesi and Papua, which, respectively, account for 0.3% and 26.7% of the total peatland area in Indonesia. In the future, studies on the $CO_2$ emissions should also cover these islands in order to provide more comprehensive data on $CO_2$ emissions from Indonesia's peatlands.

These 112 observations of primary studies, which were conducted at various sites, were further classified based on three definitions of land cover: land use as defined in [61], land cover as defined by the MoEF [62,63], and land cover as defined by the IPCC for wetlands [28]. When classifying the sites based on both land use as defined by Prananto et al. [61], these 112 studies were divided into four categories in each definition, with plantations and forests accounting for the highest number of total $CO_2$ observations for land use [61] and land cover as defined by IPCC for wetlands [28], respectively. When using the land cover classification of the MoEF, the studies were divided into nine categories with estate crops accounting for the highest number of total $CO_2$ observations. Categorizing the sites using the IPCC wetlands [28] definition resulted in six classifications, with forestland and cleared forestland, drained, as the category with the highest number of total $CO_2$ observations. There were limited observations for the MoEF bare ground and mixed dry agriculture classification, implying that more studies are needed to estimate total $CO_2$ and heterotrophic emissions for these land cover classes.

In order to fill the data gap for heterotrophic emissions, we calculated the ratio of heterotrophic emissions to total emissions from paired observations. The data for heterotrophic respiration were distributed across Aceh, Jambi, Central Kalimantan, and Riau, which represent the extent of peatlands across Indonesia. We found that the ratio of heterotrophic to total $CO_2$ emissions from paired data based on the primary studies was 78%, as depicted in Table 2.

**Table 2.** Ratio of heterotrophic to total $CO_2$ emissions from the primary studies in peatlands.

| Land Use | Number of Observations | Ratio of Heterotrophic to Total $CO_2$ Emissions |
|---|---|---|
| Cropland and fallow, drained | 1 | 0.90 |
| Forestland and cleared forestland, drained | 1 | 0.78 |
| Plantations, drained, oil palm | 14 | 0.67 |
| Plantations, drained, short rotations | 8 | 0.96 |
| Total | 24 | 0.78 |

### 3.2. Emissions Factor of CO$_2$ Emissions from Tropical Peatland in Indonesia

The random-effects models provided tier 2 estimates of the mean, standard error (SE), and 95% confidence interval (95% CI) of the total CO$_2$ emissions for all peatlands and each class of land use or land cover, as depicted in Figure 2. Using all observation data ($n = 112$), the random-effects model estimated a total CO$_2$ emissions of 48.22 Mg CO$_2$ ha$^{-1}$ year$^{-1}$ (95% CI: 42.36–54.08 Mg CO$_2$ ha$^{-1}$ year$^{-1}$) for the peatlands in Sumatra and Kalimantan. This tier 2 estimate had a lower SE of 2.96 Mg CO$_2$ ha$^{-1}$ year$^{-1}$, which was attributed to the large amount of observation data. The heterogeneity of the estimate was high ($I^2 = 95.5\%$) but statistically significant ($Q = 6940$, *p*-value < 0.01), indicating that total CO$_2$ emissions were considerably different among the study sites. Classifying the study sites into relevant land use and land cover classes produced specific estimates of total CO$_2$ emission factors, which were lower or higher than the tier 2 mean estimates. The heterogeneity of total CO$_2$ emissions between study sites within a land use/cover class was also high ($I^2$: 88.3–97.3% for land use classes, $I^2$: 40.1–97.6% for land cover classes). These findings confirmed that total CO$_2$ emissions from peatlands varied across the study sites within a particular class of land use/land cover due to variability in the environmental parameters.

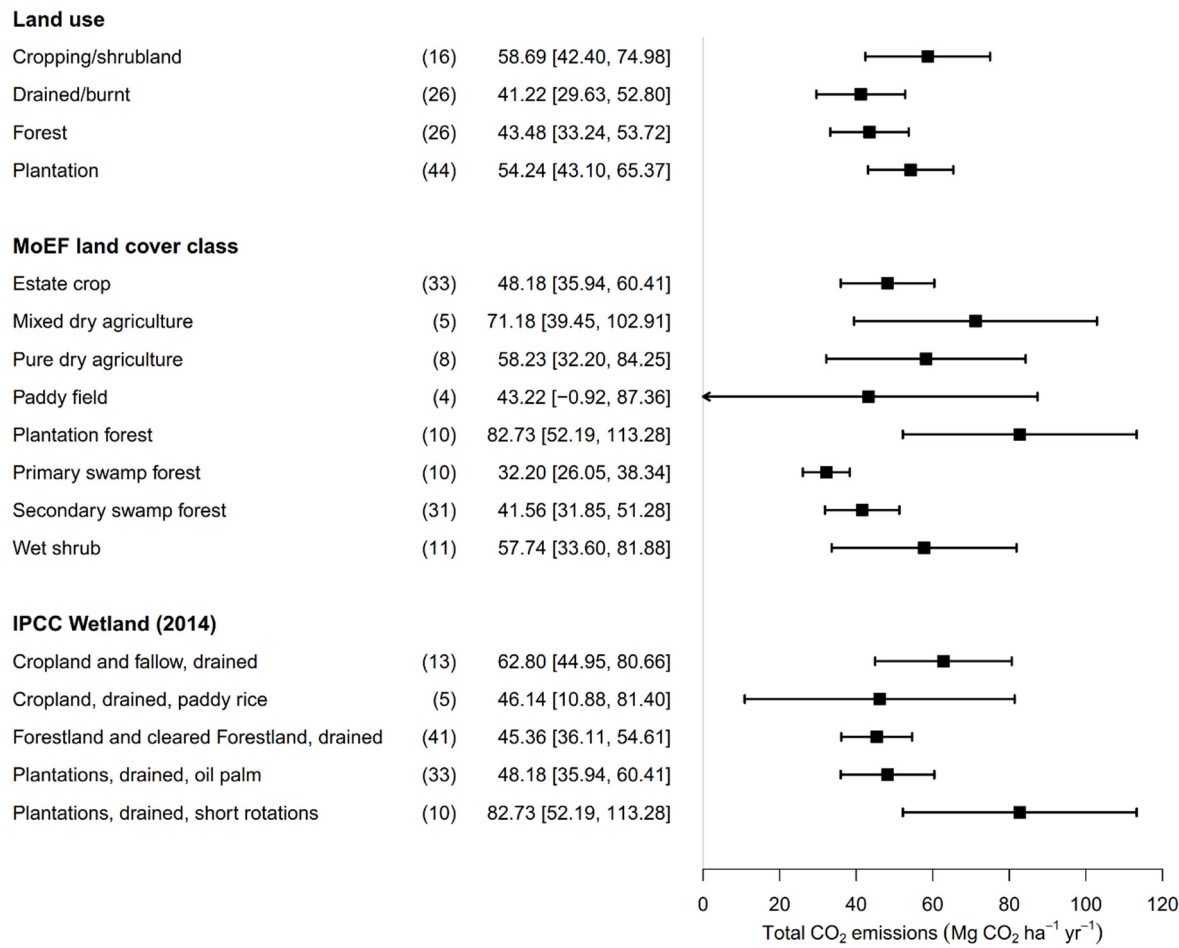

**Figure 2.** Total CO$_2$ emissions factors of Indonesia' peatlands for each land use and land cover class. The number of observations is provided in parentheses, followed by the mean and lower and upper bounds of the confidence interval in square brackets, separated by a comma.

Based on land use classification, the total CO$_2$ emissions range from 41.22 to 58.69 Mg CO$_2$ ha$^{-1}$ year$^{-1}$, with the lowest value observed in the drained/burnt class. If land cover classes as defined by the MoEF [63] and IPCC [28] are applied, the highest CO$_2$ emissions can then be observed in the plantation forest or plantation, drained, short plantation categories. Oil palm plantations (defined as estate crops based on the MoEF's land cover

class or as plantations, drained, oil palm according to the IPCC [28] had average $CO_2$ emissions of 48.18 Mg $CO_2$ ha$^{-1}$ year$^{-1}$

Due to data limitations for heterotrophic emissions, they only accounted for 31 of the 112 total measurements. Similar to the total $CO_2$ emission, the heterogeneity of heterotrophic emissions was also high ($I^2$ = 95.4%) and significant (Q = 6948, *p*-value < 0.01). The specific estimates of heterotrophic emissions for each land use or land cover type are shown in Figure 3. The results show that the average emission factor from all land use and land cover types was 38.17 Mg $CO_2$ ha$^{-1}$ year$^{-1}$, with a 95% CI of 33.63–42.71 Mg $CO_2$ ha$^{-1}$ year$^{-1}$. These numbers can be considered as the emission factor of heterotrophic emissions from each land cover/land use class at the country level. The heterogeneity of heterotrophic emissions was also high, both within the land use classes ($I^2$ = 88.3–97.3%) and within the land cover classes ($I^2$ = 40.6–97.6%), indicating that the heterotrophic emissions varied across the peatland sites, similar to the total $CO_2$ emissions.

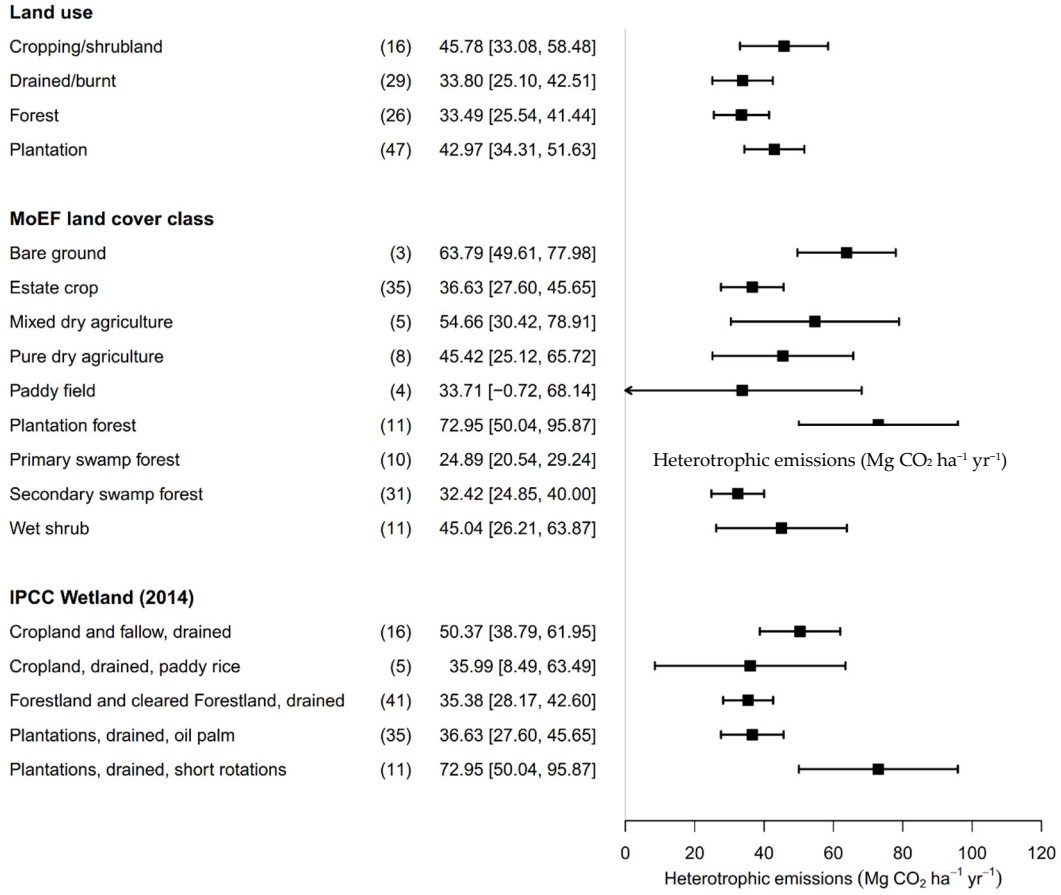

**Figure 3.** Heterotrophic emissions of Indonesia' peatlands. For each land use and land cover class, the number of observations is provided in parentheses, followed by the mean and lower and upper bounds of the confidence interval in square brackets, separated by a comma.

### 3.3. Meta-Analysis of $CO_2$ Emissions with Environmental Variables

We provide three alternative models to estimate total soil $CO_2$ emissions and other significant parameters (Table 3), including absolute latitude (*L*), water table depth (*W*), and bulk density (*B*). Temperature (*T*) and rainfall (*R*) were observed to be insignificant predictors; therefore, they were omitted from the model selection. Total soil emissions 1 (TE1) was developed using three parameters (*L*, *W*, and *B*) with an alpha of 10%; the TE1 model is a good option to estimate the total $CO_2$ emissions when the bulk density data are available. In the absence of field bulk density data, future studies can consider using the average bulk density data for the land use and land cover categories that were

collected in this study (see Table A2). Total soil emissions 2 (TE2) was used to predict total soil emissions (Mg $CO_2$ $ha^{-1}$ $year^{-1}$) using *W* and *L*. Total soil emissions 3 (TE3) was the simplest model, using *W* as the only independent predictor, but this model had the lowest $R^2$. Compared to TE1, the TE2 and TE3 models provide practical advantages for estimating the total $CO_2$ emissions, as *W* and *L* data are easy to monitor in the field. TE refers to total $CO_2$ emissions while HE refers to heterotrophic emissions.

**Table 3.** Parameter estimates and goodness of fit statistics of the general mixed-effects models for estimating total $CO_2$ emissions for all land uses.

| Model | | Parameter | | | SE | *n* | F | | $R^2$ (%) | AICc |
|---|---|---|---|---|---|---|---|---|---|---|
| TE1 | - | $\beta_0$ | 41.105 | *** | 11.394 | 60 | 12.44 | *** | 45.07 | 574.21 |
| | *W* | $\beta_1$ | 0.565 | *** | 0.131 | | | | | |
| | *L* | $\beta_2$ | −13.494 | *** | 3.530 | | | | | |
| | *B* | $\beta_3$ | 69.187 | * | 37.552 | | | | | |
| TE2 | - | $\beta_0$ | 56.738 | *** | 8.057 | 101 | 8.57 | *** | 12.32 | 988.85 |
| | *W* | $\beta_1$ | 0.245 | ** | 0.103 | | | | | |
| | *L* | $\beta_2$ | −9.147 | *** | 2.828 | | | | | |
| TE3 | - | $\beta_0$ | 38.021 | *** | 5.859 | 101 | 6.27 | ** | 4.06 | 995.91 |
| | *W* | $\beta_1$ | 0.269 | ** | 0.108 | | | | | |

*** = highly significant at 1%, ** = significant at 5%, * = significant at 10%, *n* = number of observations used in the models, SE = standard error of the parameter estimates, $R^2$ = amount of heterogeneity accounted for by the models, AICc = corrected Akaike Information Criterion.

When developing suitable models for heterotrophic emissions, we found that only water table depth and latitude were significant predictors when using an alpha of 5% and 10%, as shown in Table 4. Inclusion of latitude and water table depth predictors was preferable ($R^2$ = 16.81%), rather than only using water table depth as a predictor ($R^2$ = 5.29%), as the latter only explained 5% of the heterogeneity of heterotrophic emissions. This meant that an increase in bulk density would not significantly increase the average heterotrophic emissions. HE1 was developed to describe the relationship between heterotrophic emissions (Mg $CO_2$ $ha^{-1}$ $year^{-1}$) from *W* (water table level in cm) and *L* (absolute latitude). Using only *W* as a parameter to estimate heterotrophic emissions, HE2 had a lower $R_2$ compared to HE1. Therefore, HE1 was preferred to HE2 not only because of a higher $R^2$, but also because *W* and *L* were easily collected.

**Table 4.** Parameter estimates and goodness of fit statistics of the general mixed-effect models for estimating heterotrophic emissions for all land uses (*n* = 107).

| Model | | Parameter | | | SE | F | | $R^2$ (%) | AICc |
|---|---|---|---|---|---|---|---|---|---|
| HE1 | - | $\beta_0$ | 46.451 | *** | 6.047 | 13.31 | *** | 16.81 | 990.85 |
| | *W* | $\beta_1$ | 0.201 | ** | 0.077 | | | | |
| | *L* | $\beta_2$ | −8.587 | *** | 2.065 | | | | |
| HE2 | - | $\beta_0$ | 28.547 | *** | 4.568 | 8.4 | *** | 5.29 | 1003.63 |
| | *W* | $\beta_1$ | 0.24 | *** | 0.083 | | | | |

*** = highly significant at 1%, ** = significant at 5%, *n* = number of observations used in the models, SE = standard error of the parameter estimates, $R^2$ = amount of heterogeneity accounted for by the models, AICc = corrected Akaike Information Criterion.

This study revealed that the water table depth was positively associated with the heterogeneity of either total $CO_2$ or heterotrophic emissions. Based on the regression slopes ($\beta_1$) of the simplest models (TE3 and HE2), which were 0.27 for total $CO_2$ emissions and 0.24 for heterotrophic emissions, an increase in the water table depth by 10 cm would result in an increase in the average total $CO_2$ emissions by 2.7 Mg $CO_2$ $ha^{-1}$ $year^{-1}$ and the average heterotrophic emissions by 2.4 Mg $CO_2$ $ha^{-1}$ $year^{-1}$. The water table depth effect of 2.7 Mg $CO_2$ $ha^{-1}$ $year^{-1}$ of total $CO_2$ emissions is comparable to that found by Hooijer et al. [22], who reported an equivalent total $CO_2$ emission for burnt peatland of 3.4 Mg $CO_2$ $ha^{-1}$ $year^{-1}$ when water table depth increased by 10 cm. At the water ta-

ble depth of 70 cm, the TE3 model estimated total $CO_2$ emissions of 57 Mg $CO_2$ ha$^{-1}$ year$^{-1}$ (95% CI of 49–65 Mg $CO_2$ ha$^{-1}$ year$^{-1}$, Figure 4a) and heterotrophic emissions of 45 Mg $CO_2$ ha$^{-1}$ year$^{-1}$ (95% CI of 39–51 Mg $CO_2$ ha$^{-1}$ year$^{-1}$, Figure 4b). The estimates of total and heterotrophic emissions from this study were lower than those reported by Carlson et al. [25], who estimated total $CO_2$ emissions of 73 Mg $CO_2$ ha$^{-1}$ year$^{-1}$ (95% CI of 62–88 Mg $CO_2$ ha$^{-1}$ year$^{-1}$) and heterotrophic emissions of 62 Mg $CO_2$ ha$^{-1}$ year$^{-1}$ (95% CI of 51–73 Mg $CO_2$ ha$^{-1}$ year$^{-1}$) from peatland plantations at 70 cm of water table depth. This discrepancy is reasonable since this study used $CO_2$ emissions data from various sites across different land use classes, as depicted in Figure 4, and was not limited to plantation sites, as in the case of [25].

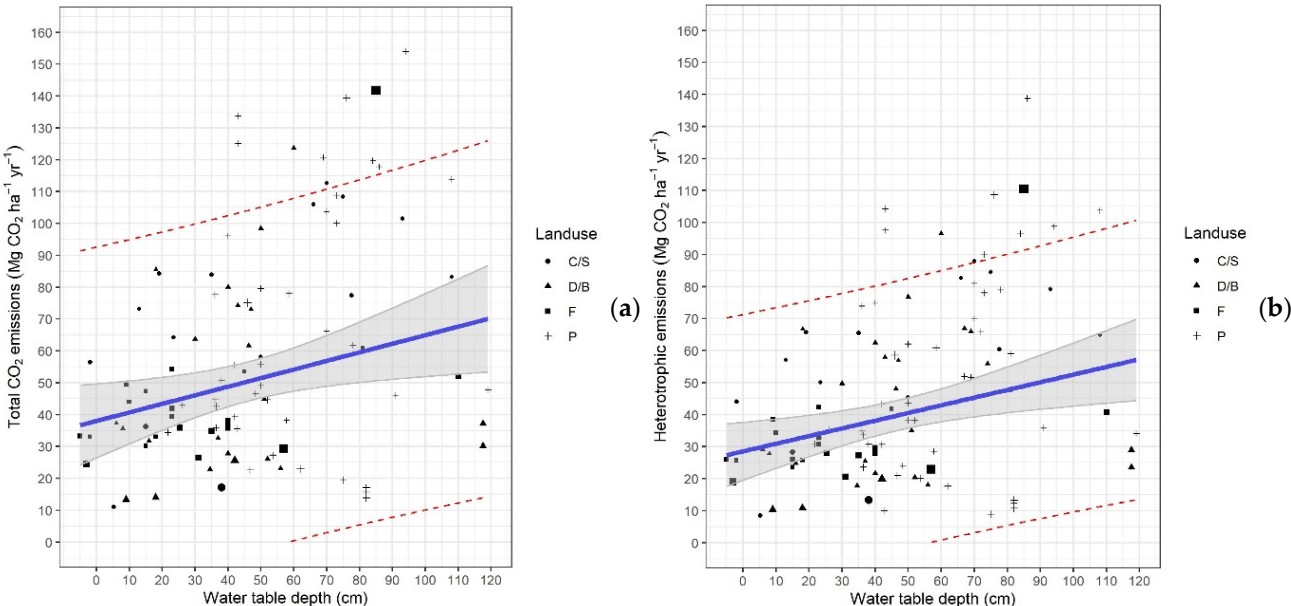

**Figure 4.** Relationship between water table depth and (**a**) total $CO_2$ emissions and (**b**) heterotrophic emissions of all land uses. Solid blue lines represent the estimates of the population mean, shaded grey bands indicate the 95% CIs of population mean estimates, dashed red lines indicate the 95% prediction intervals for the potential $CO_2$ values of future samples, while the four shapes (circles, triangles, squares, and crosses) indicate the observation values in the land use classes where C/S = cropping/shrubland, D/B = drained/burnt, F = forest, and P = plantation.

Another interesting finding of this study is that absolute latitude was a significant predictor variable in all models, suggesting that absolute latitude related well to the heterogeneity of total $CO_2$ or heterotrophic emissions. The regression coefficients of absolute latitude in all models were negative, indicating that the decrease in the absolute latitude of peatland sites resulted in an increase in the average total or heterotrophic emissions. In other words, at a given water table depth, peatland sites closer to the equator (with a latitude of 0°, Figure 5) have greater $CO_2$ emissions than those farther from the equator. For example, at a 70 cm water table depth, a peatland site located at an absolute latitude of 0.5° would have an average total $CO_2$ emissions of 69 Mg $CO_2$ ha$^{-1}$ year$^{-1}$ (95% CI of 58–80 Mg $CO_2$ ha$^{-1}$ year$^{-1}$, Figure 5a) or heterotrophic emissions of 56 Mg $CO_2$ ha$^{-1}$ year$^{-1}$ (95% CI of 48–64 Mg $CO_2$ ha$^{-1}$ year$^{-1}$, Figure 5b). These $CO_2$ emission estimates would be higher than those for a peatland site located in an absolute latitude of 3.5° at the same water table depth, which would have average total $CO_2$ emissions of 42 Mg $CO_2$ ha$^{-1}$ year$^{-1}$ (95% CI of 30–54 Mg $CO_2$ ha$^{-1}$ year$^{-1}$, Figure 5a) or heterotrophic emissions of 30 Mg $CO_2$ ha$^{-1}$ year$^{-1}$ (95% CI of 22–39 Mg $CO_2$ ha$^{-1}$ year$^{-1}$, Figure 5b).

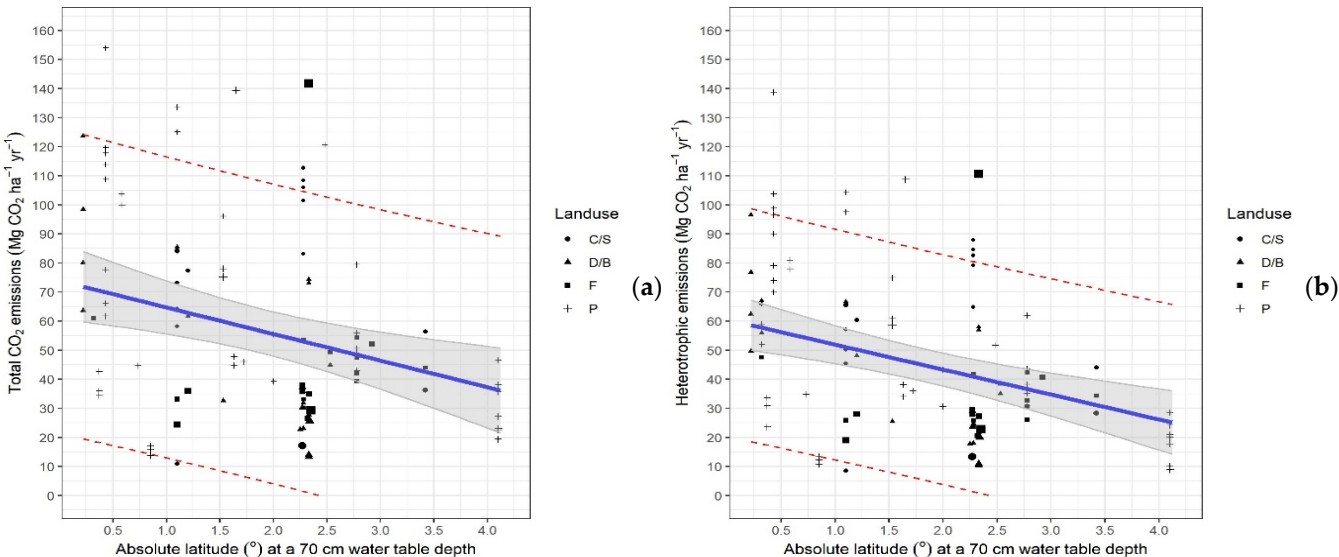

**Figure 5.** Relationship between absolute latitude at 70 cm water table depth and (**a**) total $CO_2$ emissions and (**b**) heterotrophic emissions of all land uses. Solid blue lines represent the estimates of the population mean, shaded grey bands indicate the 95% CIs of population mean estimates, dashed red lines indicate the 95% prediction intervals for the potential $CO_2$ values of future samples, while the four shapes (circles, triangles, squares, and crosses) indicate the observation values in the land use classes, where C/S = cropping/shrubland, D/B = drained/burnt, F = forest, and P = plantation.

### 3.4. Do We Need a Specific Model for Each Land Use Category?

The general mixed-effects models above provided estimates of total $CO_2$ emissions or heterotrophic emissions for all land use categories with a range of water table depths and bulk densities (see the summary statistics for water table depth and bulk density using land use classes in Table A3 and land cover classes in Table A4. To further clarify this issue, this study extended the mixed-effects models to include land use classes as dummy variables in the specific mixed-effects models, which could be used to estimate total $CO_2$ emissions for each land use class adapted from Prananto et al. [61] (CS = cropping/shrubland, DB = drained or burnt, F = primary and secondary forest, and P = oil palm, rubber and acacia plantations). Using significant predictors (*W*, *L*, and *B*) from the previous TE1, TE2, and T3 models, inclusion of land use classes as predictor variables increased the $R^2$ values only by up to 2% (Table 5).

To use the specific mixed-effects models to estimate total $CO_2$ at a particular land use, the other dummy variables (i.e., land use classes) were assumed to have zero effects. For example, the total $CO_2$ emissions of each land use class could be estimated based on the TE3-LU model as follows:

$Y = 49.950 + 0.271\ W − 17.585\ DB − 14.706\ F − 10.6096\ P$

Land use CS: $Y = 49.950 + 0.271\ W$

Land use DB: $Y_{DB} = 49.950 + 0.271\ W − 17.585\ DB$

Land use F: $Y_F = 49.950 + 0.271\ W − 14.706\ F$

Land use P: $Y_P = 49.950 + 0.271\ W − 10.6096\ P$

Using a similar approach for the total soil $CO_2$ emissions, where table depth (*W*) and Latitude (*L*) were significant predictors, we provide several recommended models in Table 6. Compared to the previous HE1 and HE models, the inclusion of land use categories only increased the $R^2$ value by 2% and 1.4%f, respectively. Therefore, similar to the total $CO_2$ emissions, we suggest that the HE1 or HE2 models can be applied for the various land use types in Indonesia.

**Table 5.** Parameter estimates and goodness of fit statistics of the general mixed-effects models for estimating total $CO_2$ emissions for all land uses.

| Model | | Parameter | | | SE | n | F | | R² (%) | AICc |
|---|---|---|---|---|---|---|---|---|---|---|
| TE1-LU | - | $\beta_0$ | 59.139 | ** | 30.076 | 60 | 6.41 | *** | 47.71 | 580.3 |
| | W | $\beta_1$ | 0.478 | *** | 0.153 | | | | | |
| | L | $\beta_2$ | −11.599 | *** | 4.19 | | | | | |
| | B | $\beta_3$ | 37.758 | | 50.016 | | | | | |
| | DB | $\beta_4$ | −14.608 | | 22.303 | | | | | |
| | F | $\beta_5$ | −17.556 | | 23.719 | | | | | |
| | P | $\beta_6$ | −5.955 | | 20.555 | | | | | |
| TE2-LU | - | $\beta_0$ | 67.808 | *** | 10.852 | 101 | 4.05 | *** | 14.49 | 992.71 |
| | W | $\beta_1$ | 0.258 | ** | 0.111 | | | | | |
| | L | $\beta_2$ | −9.287 | *** | 2.851 | | | | | |
| | DB | $\beta_3$ | −18.55 | | 10.588 | | | | | |
| | F | $\beta_4$ | −11.864 | | 10.27 | | | | | |
| | P | $\beta_5$ | −10.431 | | 10.203 | | | | | |
| TE3-LU | - | $\beta_0$ | 49.95 | *** | 9.759 | 101 | 2.27 * | | 5.62 | 999.88 |
| | W | $\beta_1$ | 0.271 | ** | 0.117 | | | | | |
| | DB | $\beta_2$ | −17.585 | | 11.079 | | | | | |
| | F | $\beta_3$ | −14.706 | | 10.74 | | | | | |
| | P | $\beta_4$ | −10.609 | | 10.661 | | | | | |

*** = highly significant at 1%, ** = significant at 5%, * = significant at 10%, *n* = number of observations used in the models, SE = standard error of the parameter estimates, R² = amount of heterogeneity accounted for by the models, AICc = corrected Akaike Information Criterion.

**Table 6.** Parameter estimates and goodness of fit statistics of the specific mixed-effects models for estimating heterotrophic emissions for each land use class.

| Model | | Parameter | | | SE | n | F | | R² (%) | AICc |
|---|---|---|---|---|---|---|---|---|---|---|
| HE1-LU | - | $\beta_0$ | 55.176 | *** | 8.259 | 107 | 5.93 | *** | 18.81 | 994.74 |
| | W | $\beta_1$ | 0.218 | *** | 0.084 | | | | | |
| | L | $\beta_2$ | −8.758 | *** | 2.088 | | | | | |
| | DB | $\beta_3$ | −14.210 | * | 8.060 | | | | | |
| | F | $\beta_4$ | −9.104 | | 7.852 | | | | | |
| | P | $\beta_5$ | −8.979 | | 7.758 | | | | | |
| HE2-LU | - | $\beta_0$ | 37.908 | *** | 7.678 | 107 | 2.71 | ** | 6.65 | 1007.84 |
| | W | $\beta_1$ | 0.244 | *** | 0.091 | | | | | |
| | DB | $\beta_2$ | −12.658 | | 8.660 | | | | | |
| | F | $\beta_3$ | −11.902 | | 8.450 | | | | | |
| | P | $\beta_4$ | −8.968 | | 8.331 | | | | | |

*** = highly significant at 1%, ** = significant at 5%, * = significant at 10%, *n* = number of observations used in the models, SE = standard error of the parameter estimates, R² = amount of heterogeneity accounted for by the models, AICc = corrected Akaike Information Criterion.

## 4. Discussions

### 4.1. Recommendation for Tier 2 Emission Factor

Based on our analysis, we recommend emission factor values of peat decomposition based on various land uses and land cover types in Indonesia because each class has specific environmental characteristics. In addition, this land cover classification is used as a basis for national forest monitoring system and REDD+ projects in Indonesia. Based on Indonesia's first FREL [62], the emission factor values of peat decomposition can be calculated based on Table 2.1 in the IPCC Wetlands Supplement (2014) [28]. This study provides recommendations to improve the emission factor values for peat decomposition, since Indonesia is currently revising its second FREL. Our mean value was within the range of total soil respiration from three different ecosystems, namely forests, sago, and palm oil (14–171 Mg $CO_2$ year$^{-1}$), located in Sarawak, Malaysia, obtained by Melling et al., (2005) [64]. Our estimate was lower than a review result for tropical peatlands in the work by Hatano et al., (2019) [65], who reported 27 and 47 Mg $CO_2$ year$^{-1}$ of mean total $CO_2$ emissions for unfertilized and fertilized areas, respectively. Hatano et al., (2019) [65] used a smaller number of observations (42 datasets) and the data were distributed across not only Indonesia, but also Malaysia.

*4.2. Estimating $CO_2$ Emissions from Water Table Depth and Latitude*

Absolute latitude is a significant predictor variable in the model because sites near the equator may have higher oscillating temperatures between day and night than those further from equator. Hoyt et al. [26] have explained that oscillating temperatures may push fluxes of $CO_2$ from the peat surface to the air through a gas transport mechanism. As the peat warms during daytime, soil gas expands, which drives a higher gas flux from the peat surface to the atmosphere on sites near the equator. This finding suggests that spatial variability of peatland sites should also be considered when managing tropical peatlands in Indonesia.

The model from this study can be applied to estimate the effects of peat rewetting on total and heterotrophic $CO_2$ emissions in Indonesia. We did not include bulk density, air temperature, or rainfall in the models because these parameters were not found to be significant predictors to estimate heterotrophic emissions. This suggests that an increase in bulk density does not significantly increase the average heterotrophic emissions. The use of bulk density data for a predictor variable is also not practical because lab measurement is needed to obtain this data. Unlike land use and land cover as categorical variables, the data for other continuous variables (i.e., water table depth, air temperature, annual rainfall, and bulk density) were not available for all primary studies. These missing data could not be inferred from the publications because the authors did not measure all of the variables used in their studies, specifically water table depth and bulk density.

Our models suggest that the significantly different $CO_2$ emissions for different land use categories are influenced more by the water table depth and latitude position for those locations relative to other observed parameters, such as bulk density, air temperature, and rainfall. The three models described in this study still represented only part of the variation in the total (4.1–45.1%) and heterotrophic (5.3–16.8%) $CO_2$ emissions, suggesting that there are other environmental variables that need to be included in future studies. Kardol and Wardle (2010) [66] have suggested that aboveground and belowground linkages, such as composition of plants and soil microbes, may contribute to the functioning of ecosystems in terms of carbon sequestration and emission. These are relevant variables to be included in the modeling of $CO_2$ emissions from peatlands in the future.

The use of an extensive dataset for the model development in this study may have resulted in better estimates of emission reduction potential from peat rewetting in Indonesia. Water table management is one of the most important strategies in peatland restoration. Hence, restoring the hydrological function of degraded peat ecosystems is key to successful revegetation, reducing fire risks, and reducing the potential $CO_2$ emissions associated with peat oxidation [17,22,67–69]. Several studies have reported that peat rewetting, which generally consists of canal blocking and canal infilling, can increase the groundwater table, hence reducing $CO_2$ emissions [13,24,67–69]. However, the number of observations remains limited.

Our study revealed that predictors such as water table depth and latitude were positively associated with the heterogeneity of either total or heterotrophic $CO_2$ emissions. This finding suggests that the spatial variability of peatland sites should also be considered, along with the water table depth, when reducing the $CO_2$ emissions from tropical peatlands in Indonesia. This study also confirmed that there is no impact from the land use category on the total and heterotrophic emissions. Further, similar to Carlson et al. [25], this result confirmed that land use classes do not actually influence the average total $CO_2$ emissions.

## 5. Conclusions

The study was conducted to apply the tier 2 emission factor for peat decomposition to recent publications in Indonesia and to model the relationship between total and heterotrophic respiration with significant environmental predictors (i.e., land use, land cover class, geographical coordinate, water table depth, bulk density, and air temperature). Our study revealed that predictors such as water table depth and latitude were positively associated with the heterogeneity of either total $CO_2$ or heterotrophic emissions. The

random-effects models provided tier 2 estimates of the mean, standard error (SE), and 95% confidence interval (95% CI) of the total $CO_2$ emissions for all peatlands and each class of the land use or land cover. Using all observation data ($n$ = 112) the random-effects model estimated total $CO_2$ emissions of 48.22 Mg $CO_2$ ha$^{-1}$ year$^{-1}$ (95% CI: 42.36–54.08 Mg $CO_2$ ha$^{-1}$ year$^{-1}$) for the peatlands in Sumatra and Kalimantan. At a given water table depth, peatland sites closer to the equator (with a latitude of 0°) have greater $CO_2$ emissions than those farther from the equator. This finding suggests that the spatial variability of peatland could influence soil $CO_2$ emission and this variable should be considered when managing and restoring degraded tropical peatlands in Indonesia. While land use and land cover categories do not necessarily affect the total $CO_2$ and heterotrophic emissions, the water table depth and latitude position are directly linked within the $CO_2$ emission dynamic. Given the limitations of the heterotrophic data in this study, further research is needed to improve our understanding of the relative contribution of heterotrophic and autotrophic emissions under different systems of peatland management.

**Author Contributions:** N.N. contributed to conceptualization, supervision, writing—original draft preparation, and funding acquisition; N.S.L. and M.L. contributed to writing—original draft preparation; T.T. contributed to the methodology, software, validation, writing—original draft preparation, and visualization; I.B. and J.J. contributed to writing—review and editing of the manuscript. All authors have read and agreed to the published version of the manuscript.

**Funding:** This research was funded by NORAD. The APC was funded by NORAD under grant number grant number GLO-4251 QZA-16/0172.

**Data Availability Statement:** Data sharing not applicable.

**Acknowledgments:** Feedbacks from Ivan Titaley are gratefully acknowledged.

**Conflicts of Interest:** The authors declare no conflict of interest. The funders had no role in the design of the study; in the collection, analyses, or interpretation of data; in the writing of the manuscript, or in the decision to publish the results.

## Appendix A

**Table A1.** Systematic Review.

| No | Citation | Literature | Title of Publication |
|----|----------|------------|----------------------|
| 1 | Ali et al., 2006 | Journal | Effect of environmental variations on $CO_2$ Efflux from a tropical peatlands in eastern Sumatera |
| 2 | Astiani et al., 2018 | Journal | Soil $CO_2$ respiration along annual crops or land-cover type gradients on West Kalimantan degraded peatland forest |
| 3 | Batubara et al., 2019 | Journal | Impact of soil collar insertion depth on microbial respiration measurements from tropical peat under an oil palm plantation |
| 4 | Comeau et al., 2016 | Journal | How do the heterotrophic and the total soil respiration of an oil palm plantation on peat respond to nitrogen fertilizer application? |
| 5 | Dariah et al., 2013 | Journal | Root and peat based $CO_2$ emissions from oil palm plantations |
| 6 | Furukawa et al., 2005 | Journal | Effect of changing groundwater levels caused by land-use changes on greenhouse gas fluxes from tropical peat lands |

**Table A1.** *Cont*.

| No | Citation | Literature | Title of Publication |
|----|----------|-----------|----------------------|
| 7 | Hadi et al., 2005 | Journal | Greenhouse gas emissions from tropical peatlands of Kalimantan, Indonesia |
| 8 | Handayani et al., 2009 | Journal | Carbon Dioxide ($CO_2$) Emission of Oil Palm Plantation on Peatland: The evaluation $CO_2$ flux on inside and outside Rhyzosphere |
| 9 | Hergoualc'h et al., 2017 | Journal | Total and heterotrophic soil respiration in a swamp forest and oil palm plantations on peat in Central Kalimantan |
| 10 | Hirano et al., 2007 | Journal | Carbon dioxide balance of a tropical peat swamp forest in Kalimantan, Indonesia |
| 11 | Hirano et al., 2009 | Journal | Controls on the Carbon Balance of Tropical Peatlands |
| 12 | Hirano et al., 2014 | Journal | Carbon dioxide emissions through oxidative peat decomposition on a burnt tropical peatland |
| 13 | Hooijer et al., 2012 | Journal | Subsidence and carbon loss in drained tropical peatlands |
| 14 | Hooijer et al., 2014 | Report | Carbon emissions from drained and degraded peatland in Indonesia and emission factors for measurement, reporting and verification (MRV) of peatland greenhouse gas emissions |
| 15 | Husnain et al., 2014 | Journal | $CO_2$ emissions from tropical drained peat in Sumatera, Indonesia |
| 16 | Inubushi et al., 2003 | Journal | Seasonal changes of $CO_2$, $CH_4$ and $N_2O$ fluxes in relation to land-use change in tropical peatlands located in coastal area of South Kalimantan |
| 17 | Ishikura et al., 2017 | Journal | Effect of groundwater level fluctuation on soil respiration rate of tropical peatland in Central Kalimantan, Indonesia |
| 18 | Ishizuka et al., 2002 | Journal | An intensive field study on $CO_2$, $CH_4$, and $N_2O$ emissions from soils at four land-use types in Sumatra, Indonesia |
| 19 | Itoh et al., 2017 | Journal | Factors affecting oxidative peat decomposition due to land use in tropical peat swamp forests in Indonesia |
| 20 | Jamaludin et al., 2020 | Journal | Emisi karbon dioksida ($CO_2$) dari pertanian skala kecil di lahan gambut |
| 21 | Jauhiainen et al., 2005 | Journal | Carbon fluxes from a tropical peat swamp forest floor |
| 22 | Jauhiainen et al., 2008 | Journal | Carbon dioxide and methane fluxes in drained tropical peat before and after hydrological restoration |
| 23 | Jauhiainen et al., 2012 | Journal | Carbon dioxide emissions from an Acacia plantation on peatland in Sumatera, Indonesia |

**Table A1.** *Cont.*

| No | Citation | Literature | Title of Publication |
|----|----------|------------|---------------------|
| 24 | Khasanah & Noordwijk, 2017 | Journal | Subsidence and carbon dioxide emissions in a smallholder peatland mosaic in Sumatra, Indonesia |
| 25 | Marwanto & Agus, 2013 | Journal | Is $CO_2$ flux from oil palm plantations on peatland controlled by soil moisture and/or soil and air temperatures |
| 26 | Marwanto et al., 2019 | Journal | Importance of $CO_2$ production in subsoil layers of drained tropical peatland under mature oil palm plantation |
| 27 | Saragi-Sasmito et al., 2018 | Journal | Carbon stocks, emissions, and aboveground productivity in restored secondary tropical peat swamp forest |
| 28 | Sundari et al., 2012 | Journal | Effect of groundwater level on soil respiration in tropical peat swamp forests |
| 29 | Swails et al., 2018 | Journal | The response of soil respiration to climatic drivers in undrained forest and drained oil palm plantations in a Indonesia peatland |
| 30 | Toma et al., 2011 | Journal | Nitrous oxide emission derived from soil organic matter decomposition from tropical agricultural peat soil in central Kalimantan, Indonesia |
| 31 | Wakhid et al., 2017 | Journal | Soil carbon dioxide emissions from a rubber plantation on tropical peat |
| 32 | Watanabe et al., 2009 | Journal | Methane and $CO_2$ fluxes from an Indonesian peatland used for sago palm (Metroxylon sagu Rottb.) cultivation: Effects of fertilizer and groundwater level management |

**Table A2.** Datasets.

| No. | Authors | | Site Name | Land Use | Land Cover | Province | Latitude | Longitude | Total CO$_2$ Mean | SD | Heterotrophic Mean | SD | | WTD | Temp. | Rainfall | BD |
|---|---|---|---|---|---|---|---|---|---|---|---|---|---|---|---|---|---|
| 1 | Ali et al., (2006) | Site 01 | Agriculture land | C/S | Mixed dry agriculture | Jambi | −1.2 | 103.7 | 77.45 | 18.64 | 60.41 | 14.54 | | 77.5 | 28 | 2490 | 0.4 |
| 2 | Ali et al., (2006) | Site 02 | Logged forest | F | Secondary swamp forest | Jambi | −1.2 | 103.7 | 35.95 | 3.05 | 28.04 | 2.38 | | 25.3 | 26.8 | 2490 | 0.28 |
| 3 | Ali et al., (2006) | Site 03 | Recently burned | D/B | Secondary swamp forest | Jambi | −1.2 | 103.7 | 61.63 | 16.11 | 48.07 | 12.57 | | 46.3 | 27.8 | 2490 | 0.32 |
| 4 | Astiani et al., (2018) | Site 01 | Drained and cleared swamp forest | D/B | Wet shrub | West Kalimantan | −0.22 | 109.43 | 63.7 | 10.2 | 49.69 | 7.96 | | 30 | 26.5 | 3171 | 0.15 |
| 5 | Astiani et al., (2018) | Site 02 | Drained and cleared swamp forest | D/B | Wet shrub | West Kalimantan | −0.22 | 109.43 | 80.1 | 11.8 | 62.48 | 9.2 | | 40 | 26.5 | 3171 | 0.15 |
| 6 | Astiani et al., (2018) | Site 03 | Drained and cleared swamp forest | D/B | Wet shrub | West Kalimantan | −0.22 | 109.43 | 98.5 | 11.9 | 76.83 | 9.28 | | 50 | 26.5 | 3171 | 0.15 |
| 7 | Astiani et al., (2018) | Site 04 | Drained and cleared swamp forest | D/B | Wet shrub | West Kalimantan | −0.22 | 109.43 | 123.7 | 12.5 | 96.49 | 9.75 | | 60 | 26.5 | 3171 | 0.15 |
| 8 | Batubara et al., (2019) | Site 01 | Oil palm plantation | P | Estate crop | North Sumatra | 2 | 100.27 | 39.3 | 18.67 | 30.65 | 14.56 | | 42 | 26 | 3467 | 0.15 |
| 9 | Batubara et al., (2019) | Site 02 | Oil palm plantation | P | Estate crop | North Sumatra | 2 | 100.27 | 55.5 | 22.06 | 43.29 | 17.21 | | 42 | 26 | 3467 | 0.15 |
| 10 | Comeau et al., (2016) | Site 01 | Oil palm plantation | P | Estate crop | Jambi | −1.65 | 103.87 | 139.4 | 5.66 | 108.73 | 4.41 | | 76 | 27.5 | 2466 | 0.23 |
| 11 | Dariah et al., (2014) | Site 01 | Oil palm plantation | P | Estate crop | Jambi | −1.63 | 103.78 | 44.7 | 11.12 | 38.2 | 9.47 | a | 52 | 26.2 | 2500 | 0.16 |
| 12 | Dariah et al., (2014) | Site 02 | Oil palm plantation | P | Estate crop | Jambi | −1.63 | 103.78 | 47.8 | 21.36 | 34.1 | 15.84 | a | 119 | 26.2 | 2500 | 0.19 |
| 13 | Furukawa et al., (2005) | Site 01 | Cassava field | C/S | Pure dry agriculture | Jambi | −1.1 | 103.71 | 64.3 | 32.04 | 50.15 | 24.99 | | 23.5 | 26.7 | 2582 | na |
| 14 | Furukawa et al., (2005) | Site 02 | Coconut field | P | Estate crop | Jambi | −1.1 | 103.71 | 133.7 | 36.55 | 104.29 | 28.51 | | 43 | 26.7 | 2582 | na |
| 15 | Furukawa et al., (2005) | Site 03 | Coconut field | P | Estate crop | Jambi | −1.1 | 103.71 | 125.1 | 31.99 | 97.58 | 24.95 | | 43 | 26.7 | 2582 | na |
| 16 | Furukawa et al., (2005) | Site 04 | Drained forest | D/B | Secondary swamp forest | Jambi | −1.1 | 103.71 | 85.54 | 37.38 | 66.72 | 29.16 | | 18.1 | 26.7 | 2582 | na |
| 17 | Furukawa et al., (2005) | Site 05 | Lowland paddy field | C/S | Paddy field | Jambi | −1.1 | 103.71 | 11.05 | 11.47 | 8.62 | 8.95 | | 5.2 | 26.7 | 2582 | na |
| 18 | Furukawa et al., (2005) | Site 06 | Pineapple field | C/S | Pure dry agriculture | Jambi | −1.1 | 103.71 | 84.38 | 10.51 | 65.82 | 8.2 | | 19 | 26.7 | 2582 | na |
| 19 | Furukawa et al., (2005) | Site 07 | Pineapple field | C/S | Pure dry agriculture | Jambi | −1.1 | 103.71 | 84.03 | 18.03 | 65.54 | 14.06 | | 35 | 26.7 | 2582 | na |
| 20 | Furukawa et al., (2005) | Site 08 | Pineapple field | C/S | Pure dry agriculture | Jambi | −1.1 | 103.71 | 58.2 | 22.36 | 45.4 | 17.44 | | 50 | 26.7 | 2582 | na |
| 21 | Furukawa et al., (2005) | Site 09 | Swampy forest | F | Secondary swamp forest | Jambi | −1.1 | 103.71 | 33.31 | 8.63 | 25.98 | 6.73 | | −5 | 26.7 | 2582 | na |
| 22 | Furukawa et al., (2005) | Site 10 | Swampy forest | F | Secondary swamp forest | Jambi | −1.1 | 103.71 | 24.41 | 2.37 | 19.04 | 1.85 | | −3 | 26.7 | 2582 | na |
| 23 | Furukawa et al., (2005) | Site 11 | Swampy forest | F | Secondary swamp forest | Jambi | −1.1 | 103.71 | 33.02 | 16.3 | 25.76 | 12.71 | | −2 | 26.7 | 2582 | na |

**Table A2.** *Cont.*

| No. | Authors | | Site Name | Land Use | Land Cover | Province | Latitude | Longitude | Total CO$_2$ Mean | SD | Heterotrophic Mean | SD | | WTD | Temp. | Rainfall | BD |
|---|---|---|---|---|---|---|---|---|---|---|---|---|---|---|---|---|---|
| 24 | Furukawa et al., (2005) | Site 12 | Upland paddy field | C/S | Paddy field | Jambi | −1.1 | 103.71 | 73.2 | 34.71 | 57.1 | 27.07 | | 13 | 26.7 | 2582 | na |
| 25 | Hadi et al., (2005) | Site 01 | Abandoned paddy-crop field | F | Secondary swamp forest | South Kaliman-tan | −2.37 | 115.37 | 87.63 | 29.12 | 68.35 | 22.71 | | na | 26.5 | 2756 | na |
| 26 | Hadi et al., (2005) | Site 02 | Paddy field | C/S | Paddy field | South Kaliman-tan | −2.37 | 115.37 | 57.76 | 30.16 | 45.05 | 23.52 | | na | 26.5 | 2756 | na |
| 27 | Hadi et al., (2005) | Site 03 | Secondary forest | F | Secondary swamp forest | South Kaliman-tan | −2.37 | 115.37 | 46.05 | 25.1 | 35.92 | 19.58 | | na | 26.5 | 2756 | na |
| 28 | Handayani et al., (2009) | Site 01 | Oil palm plantation | P | Estate crop | Aceh | 4.1 | 96.21 | 22.99 | 4.94 | 17.75 | 6.4 | a | 62 | 36.2 | 2789 | na |
| 29 | Handayani et al., (2009) | Site 02 | Oil palm plantation | P | Estate crop | Aceh | 4.1 | 96.21 | 19.39 | 9.91 | 8.89 | 5.25 | a | 75 | 36.2 | 2789 | na |
| 30 | Handayani et al., (2009) | Site 03 | Oil palm plantation | P | Estate crop | Aceh | 4.1 | 96.21 | 46.57 | 23.3 | 24.1 | 26.79 | a | 48.4 | 36.2 | 2789 | na |
| 31 | Handayani et al., (2009) | Site 04 | Oil palm plantation | P | Estate crop | Aceh | 4.1 | 96.21 | 27.22 | 8.07 | 20.05 | 6.24 | a | 53.6 | 36.2 | 2789 | na |
| 32 | Handayani et al., (2009) | Site 05 | Oil palm plantation | P | Estate crop | Aceh | 4.1 | 96.21 | 38.19 | 25.16 | 28.55 | 16.97 | a | 57.8 | 36.2 | 2789 | na |
| 33 | Handayani et al., (2009) | Site 06 | Oil palm plantation | P | Estate crop | Aceh | 4.1 | 96.21 | 22.58 | 7.33 | 20.97 | 7.72 | a | 46.7 | 36.2 | 2789 | na |
| 34 | Handayani et al., (2009) | Site 07 | Oil palm plantation | P | Estate crop | Aceh | 4.1 | 96.21 | 35.59 | 25.41 | 10.04 | 7.98 | a | 42.7 | 36.2 | 2789 | na |
| 35 | Hergoualc'h et al., (2017) | Site 01 | Oil palm plantation | P | Estate crop | Central Kaliman-tan | −2.78 | 111.8 | 50.65 | 12.92 | 30.8 | 18.64 | a | 38 | 29.7 | 2058 | 0.31 |
| 36 | Hergoualc'h et al., (2017) | Site 02 | Oil palm plantation | P | Estate crop | Central Kaliman-tan | −2.78 | 111.8 | 42.94 | 30.2 | 35.23 | 10.18 | a | 26 | 36.8 | 2058 | 0.33 |
| 37 | Hergoualc'h et al., (2017) | Site 03 | Primary peat forest | F | Primary swamp forest | Central Kaliman-tan | −2.78 | 111.8 | 47.34 | 21.02 | 26.06 | 5.09 | a | 15 | 29.6 | 2058 | 0.17 |
| 38 | Hirano et al., (2007) | Site 01 | Secondary forest | F | Secondary swamp forest | Central Kaliman-tan | −2.33 | 114.04 | 141.77 | 1.33 | 110.58 | 1.04 | | 85 | 26.3 | 2235 | na |
| 39 | Hirano et al., (2009) | Site 01 | Crop-free agric land | C/S | Pure dry agriculture | Central Kaliman-tan | −2.27 | 113.98 | 17.2 | 1.6 | 13.42 | 1.25 | | 38 | 26.3 | 2331 | na |
| 40 | Hirano et al., (2009) | Site 02 | Drained regenerating forest | D/B | Secondary swamp forest | Central Kaliman-tan | −2.27 | 113.98 | 37.2 | 3.8 | 29.02 | 2.96 | | 117.5 | 26.3 | 2560 | na |
| 41 | Hirano et al., (2009) | Site 03 | Drained regenerating forest | D/B | Secondary swamp forest | Central Kaliman-tan | −2.27 | 113.98 | 30.2 | 3.6 | 23.56 | 2.81 | | 117.5 | 26.3 | 2331 | na |
| 42 | Hirano et al., (2009) | Site 04 | Secondary forest | F | Secondary swamp forest | Central Kaliman-tan | −2.27 | 113.98 | 35.7 | 5.8 | 27.85 | 4.52 | | 40 | 26.3 | 1852 | na |

**Table A2.** *Cont.*

| No. | Authors | Site Name | | Land Use | Land Cover | Province | Latitude | Longitude | Total CO$_2$ Mean | SD | Heterotrophic Mean | SD | | WTD | Temp. | Rainfall | BD |
|---|---|---|---|---|---|---|---|---|---|---|---|---|---|---|---|---|---|
| 43 | Hirano et al., (2009) | Site 05 | Secondary forest | F | Secondary swamp forest | Central Kalimantan | −2.27 | 113.98 | 37.1 | 5.2 | 28.94 | 4.06 | | 40 | 26.3 | 2292 | na |
| 44 | Hirano et al., (2009) | Site 06 | Secondary forest | F | Secondary swamp forest | Central Kalimantan | −2.27 | 113.98 | 38 | 4.9 | 29.64 | 3.82 | | 40 | 26.3 | 2560 | na |
| 45 | Hirano et al., (2014) | Site 01 | Burned forest | D/B | Secondary swamp forest | Central Kalimantan | −2.33 | 114.03 | 14 | 2.99 | 10.92 | 2.33 | | 18 | 26.2 | 2540 | na |
| 46 | Hirano et al., (2014) | Site 02 | Burned forest | D/B | Secondary swamp forest | Central Kalimantan | −2.33 | 114.03 | 13.3 | 2.69 | 10.37 | 2.1 | | 9 | 26.2 | 2540 | na |
| 47 | Hooijer et al., (2012) | Site 01 | Acacia plantation | P | Plantation forest | Riau | 0.58 | 102.33 | 103.75 | 50.3 | 80.93 | 39.23 | | 70 | 30 | 2500 | 0.09 |
| 48 | Hooijer et al., (2012) | Site 02 | Oil palm plantation | P | Estate crop | Jambi | 0.58 | 102.33 | 100 | 17.9 | 78 | 13.96 | | 73 | 30 | 2500 | 0.09 |
| 49 | Hooijer et al., (2014) | Site 01 | Burnt and drained peatland | D/B | Secondary swamp forest | Central Kalimantan | −2.25 | 114.58 | 22.75 | 8.82 | 17.75 | 6.88 | | 34.5 | 25.9 | 2842 | 0.09 |
| 50 | Husnain et al., (2014) | Site 01 | Acacia plantation | P | Plantation forest | Riau | 0.32 | 101.68 | na | na | 59 | 19.02 | a | 81 | 31.9 | 2492 | 0.12 |
| 51 | Husnain et al., (2014) | Site 02 | Bare ground | D/B | Bare ground | Riau | 0.32 | 101.68 | na | na | 67 | 23.97 | a | 67 | 31.9 | 2492 | 0.12 |
| 52 | Husnain et al., (2014) | Site 03 | Bare ground | D/B | Bare ground | Riau | 0.32 | 101.68 | na | na | 56 | 30.06 | a | 74 | 31.9 | 2492 | 0.12 |
| 53 | Husnain et al., (2014) | Site 04 | Bare ground | D/B | Bare ground | Riau | 0.32 | 101.68 | na | na | 66 | 26.95 | a | 69 | 31.9 | 2492 | 0.12 |
| 54 | Husnain et al., (2014) | Site 05 | Oil palm plantation | P | Estate crop | Riau | 0.32 | 101.68 | na | na | 66 | 25.03 | a | 72 | 31.9 | 2492 | 0.15 |
| 55 | Husnain et al., (2014) | Site 07 | Rubber plantation | P | Estate crop | Riau | 0.32 | 101.68 | na | na | 52 | 16.97 | a | 67 | 31.9 | 2492 | 0.12 |
| 56 | Husnain et al., (2014) | Site 08 | Secondary forest | F | Secondary swamp forest | Riau | 0.32 | 101.68 | 61 | 25.03 | 47.58 | 19.52 | | 81 | 31.9 | 2492 | 0.12 |
| 57 | Inubushi et al., (2003) | Site 01 | Abandoned cropland | C/S | Pure dry agriculture | South Kalimantan | −3.42 | 114.67 | 36.3 | 4.04 | 28.31 | 3.15 | | 15 | 26.5 | 3133 | na |
| 58 | Inubushi et al., (2003) | Site 02 | Abandoned paddy | C/S | Paddy field | South Kalimantan | −3.42 | 114.67 | 56.5 | 10.63 | 44.07 | 8.29 | | −2 | 26.5 | 3133 | na |
| 59 | Inubushi et al., (2003) | Site 03 | Secondary forest | F | Secondary swamp forest | South Kalimantan | −3.42 | 114.67 | 44 | 15.76 | 34.32 | 12.29 | | 10 | 26.5 | 3133 | na |
| 60 | Ishikura et al., (2017) | Site 01 | Burned land | D/B | Wet shrub | Central Kalimantan | −2.28 | 114.01 | 31.8 | 22.24 | 24.8 | 17.35 | | 16 | 26.3 | 2235 | 0.22 |
| 61 | Ishikura et al., (2017) | Site 02 | Burned land | D/B | Wet shrub | Central Kalimantan | −2.28 | 114.01 | 23.1 | 10.61 | 18.02 | 8.28 | | 56 | 26.3 | 2235 | 0.22 |

**Table A2.** *Cont.*

| No. | Authors | Site Name | Land Use | Land Cover | Province | Latitude | Longitude | Total CO$_2$ Mean | SD | Heterotrophic Mean | SD | WTD | Temp. | Rainfall | BD |
|---|---|---|---|---|---|---|---|---|---|---|---|---|---|---|---|
| 62 | Ishikura et al., (2017) | Site 03 | Burned land | D/B | Wet shrub | Central Kaliman-tan | −2.28 | 114.01 | 37.3 | 45.88 | 29.09 | 35.79 | 6 | 26.3 | 2235 | 0.13 |
| 63 | Ishikura et al., (2017) | Site 04 | Burned land | D/B | Wet shrub | Central Kaliman-tan | −2.28 | 114.01 | 35.7 | 28.61 | 27.85 | 22.32 | 8 | 26.3 | 2235 | 0.13 |
| 64 | Ishikura et al., (2017) | Site 05 | Crop land | C/S | Pure dry agriculture | Central Kaliman-tan | −2.28 | 114.01 | 112.7 | 59.37 | 87.91 | 46.31 | 70 | 26.3 | 2235 | 0.38 |
| 65 | Ishikura et al., (2017) | Site 06 | Crop land | C/S | Mixed dry agriculture | Central Kaliman-tan | −2.28 | 114.01 | 101.5 | 55.61 | 79.17 | 43.38 | 93 | 26.3 | 2235 | 0.38 |
| 66 | Ishikura et al., (2017) | Site 07 | Crop land | C/S | Pure dry agriculture | Central Kaliman-tan | −2.28 | 114.01 | 106 | 56.17 | 82.68 | 43.81 | 66 | 26.3 | 2235 | 0.42 |
| 67 | Ishikura et al., (2017) | Site 08 | Forest land | F | Secondary swamp forest | Central Kaliman-tan | −2.28 | 114.01 | 53.6 | 20.86 | 41.81 | 16.27 | 45 | 26.3 | 2235 | 0.13 |
| 68 | Ishikura et al., (2017) | Site 09 | Forest land | F | Primary swamp forest | Central Kaliman-tan | −2.28 | 114.01 | 30.2 | 18.58 | 23.56 | 14.49 | 15 | 26.3 | 2235 | 0.12 |
| 69 | Ishikura et al., (2017) | Site 10 | Forest land | F | Primary swamp forest | Central Kaliman-tan | −2.28 | 114.01 | 33.1 | 16.07 | 25.82 | 12.53 | 18 | 26.3 | 2235 | 0.12 |
| 70 | Ishikura et al., (2017) | Site 11 | Grass land | C/S | Wet shrub | Central Kaliman-tan | −2.28 | 114.01 | 83.2 | 48.48 | 64.9 | 37.81 | 108 | 26.3 | 2235 | 0.33 |
| 71 | Ishizuka et al., (2002) | Site 01 | Deforested area | D/B | Plantation forest | Jambi | −1.05 | 102.15 | 16.48 | 8.79 | 12.85 | 6.86 | na | 25.7 | 2060 | 1.19 |
| 72 | Ishizuka et al., (2002) | Site 02 | Logged-over forest | D/B | Secondary swamp forest | Jambi | −1.05 | 102.15 | 20.97 | 5.67 | 16.36 | 4.42 | na | 25.7 | 2060 | 1.14 |
| 73 | Ishizuka et al., (2002) | Site 03 | Logged-over forest | D/B | Secondary swamp forest | Jambi | −1.05 | 102.15 | 26.05 | 7.53 | 20.32 | 5.87 | na | 25.7 | 2060 | 1.14 |
| 74 | Ishizuka et al., (2002) | Site 04 | Logged-over forest | D/B | Secondary swamp forest | Jambi | −1.05 | 102.15 | 37.26 | 12.52 | 29.06 | 9.77 | na | 25.7 | 2060 | 1.08 |
| 75 | Ishizuka et al., (2002) | Site 05 | Oil palm plantation | P | Estate crop | Jambi | −1.05 | 102.15 | 18.31 | 4.17 | 14.28 | 3.25 | na | 25.7 | 2060 | 1.18 |
| 76 | Ishizuka et al., (2002) | Site 06 | Primary forest | F | Primary swamp forest | Jambi | −1.05 | 102.15 | 20.33 | 4.95 | 15.86 | 3.86 | na | 25.7 | 2060 | 1.17 |
| 77 | Ishizuka et al., (2002) | Site 07 | Primary forest | F | Primary swamp forest | Jambi | −1.05 | 102.15 | 30.1 | 8.64 | 23.48 | 6.74 | na | 25.7 | 2060 | 1.17 |
| 78 | Ishizuka et al., (2002) | Site 08 | Rubber plantation | P | Estate crop | Jambi | −1.05 | 102.15 | 23.96 | 8.52 | 18.69 | 6.65 | na | 25.7 | 2060 | 1.12 |
| 79 | Itoh et al., (2017) | Site 01 | Drained and burnt forest | D/B | Secondary swamp forest | Central Kaliman-tan | −2.34 | 114.04 | 25.57 | 2.42 | 19.94 | 1.89 | 42 | 26.2 | 2546 | 0.24 |
| 80 | Itoh et al., (2017) | Site 02 | Drained forest | F | Secondary swamp forest | Central Kaliman-tan | −2.35 | 114.04 | 29.32 | 1.35 | 22.87 | 1.05 | 57 | 26.2 | 2546 | 0.14 |

**Table A2.** *Cont.*

| No. | Authors | Site Name | | Land Use | Land Cover | Province | Latitude | Longitude | Total CO$_2$ | | Heterotrophic | | | WTD | Temp. | Rainfall | BD |
|---|---|---|---|---|---|---|---|---|---|---|---|---|---|---|---|---|---|
| | | | | | | | | | Mean | SD | Mean | SD | | | | | |
| 81 | Itoh et al., (2017) | Site 03 | Undrained forest | F | Primary swamp forest | Central Kalimantan | −2.32 | 113.9 | 26.44 | 3.65 | 20.62 | 2.85 | | 31 | 26.2 | 2546 | 0.11 |
| 82 | Jamaludin et al. (2020) | Site 01 | Ginger | P | Mixed dry agriculture | West Kalimantan | −0.37 | 109.52 | 34.41 | 17.92 | 30.87 | 9.38 | a | 21.8 | 31.6 | na | 0.14 |
| 83 | Jamaludin et al. (2020) | Site 02 | Oil palm plantation | P | Estate crop | West Kalimantan | −0.37 | 109.52 | 35.93 | 14.17 | 23.56 | 8.15 | a | 36.4 | 30.6 | na | 0.21 |
| 84 | Jamaludin et al., (2020) | Site 03 | Rubber plantation | P | Estate crop | West Kalimantan | −0.37 | 109.52 | 42.63 | 12.57 | 33.67 | 11.85 | a | 36.4 | 30.3 | na | 0.17 |
| 85 | Jauhiainen et al., (2005) | Site 03 | Undrained peat swamp forest | F | Primary swamp forest | Central Kalimantan | −2.33 | 113.92 | 34.93 | 3.2 | 27.25 | 2.5 | | 35 | 25.5 | 2528 | 0.15 |
| 86 | Jauhiainen et al., (2008) | Site 01 | Deforested, drained, and burned peat forest | D/B | Wet shrub | Central Kalimantan | −2.33 | 114.03 | 27.81 | 8.16 | 21.69 | 6.36 | | 40 | 33.5 | 2331 | na |
| 87 | Jauhiainen et al., (2008) | Site 02 | Deforested, drained, and burned peat forest | D/B | Wet shrub | Central Kalimantan | −2.33 | 114.03 | 26.08 | 7.71 | 20.34 | 6.01 | | 52 | 33.5 | 2560 | na |
| 88 | Jauhiainen et al., (2008) | Site 03 | Drained and selectively logged peat swamp | D/B | Secondary swamp forest | Central Kalimantan | −2.33 | 114.03 | 73.05 | 39.94 | 56.98 | 31.15 | | 47 | 29.3 | 2331 | na |
| 89 | Jauhiainen et al., (2008) | Site 04 | Drained and selectively logged peat swamp | D/B | Secondary swamp forest | Central Kalimantan | −2.33 | 114.03 | 74.4 | 42.85 | 58.03 | 33.42 | | 43 | 29.3 | 2560 | na |
| 90 | Jauhiainen et al., (2012) | Site 01 | Acacia plantation | P | Plantation forest | Riau | 0.43 | 101.88 | 154 | 56.79 | 98.81 | 25.62 | a | 94 | 26.2 | 2500 | 0.11 |
| 91 | Jauhiainen et al., (2012) | Site 02 | Acacia plantation | P | Plantation forest | Riau | 0.43 | 101.88 | 108.8 | 37.3 | 90.05 | 31.88 | a | 73 | 26.2 | 2500 | 0.12 |
| 92 | Jauhiainen et al., (2012) | Site 03 | Acacia plantation | P | Plantation forest | Riau | 0.43 | 101.88 | 113.88 | 52.15 | 103.81 | 44.4 | a | 108 | 26.2 | 2500 | 0.08 |
| 93 | Jauhiainen et al., (2012) | Site 04 | Acacia plantation | P | Plantation forest | Riau | 0.43 | 101.88 | 61.76 | 27.97 | 79.1 | 22.16 | a | 78 | 26.2 | 2500 | 0.08 |
| 94 | Jauhiainen et al., (2012) | Site 05 | Acacia plantation | P | Plantation forest | Riau | 0.43 | 101.88 | 66.14 | 66.13 | 69.99 | 34.36 | a | 70 | 26.2 | 2500 | 0.07 |
| 95 | Jauhiainen et al., (2012) | Site 06 | Acacia plantation | P | Plantation forest | Riau | 0.43 | 101.88 | 119.66 | 42.07 | 96.62 | 36.48 | a | 84 | 26.2 | 2500 | 0.06 |
| 96 | Jauhiainen et al., (2012) | Site 07 | Acacia plantation | P | Plantation forest | Riau | 0.43 | 101.88 | 77.7 | 23.72 | 73.93 | 24.81 | a | 36 | 26.2 | 2500 | 0.06 |
| 97 | Jauhiainen et al., (2012) | Site 08 | Acacia plantation | P | Plantation forest | Riau | 0.43 | 101.88 | 117.82 | 34.84 | 138.76 | 43.46 | a | 86 | 26.2 | 2500 | 0.06 |
| 98 | Khasanah & Noordwijk (2018) | Site 01 | Logged-over forest | D/B | Secondary swamp forest | Jambi | −1.53 | 102.37 | 32.63 | 19.83 | 25.45 | 15.47 | | 37 | 30 | 2349 | 0.12 |
| 99 | Khasanah & Noordwijk (2018) | Site 02 | Mixed betel nut, coconut and coffee | P | Mixed dry agriculture | Jambi | −1.53 | 102.37 | 78 | 9.9 | 60.84 | 7.72 | | 58.5 | 30 | 2349 | 0.17 |

**Table A2.** *Cont.*

| No. | Authors | Site Name | Land Use | Land Cover | Province | Latitude | Longitude | Total CO$_2$ Mean | SD | Heterotrophic Mean | SD | | WTD | Temp. | Rainfall | BD |
|---|---|---|---|---|---|---|---|---|---|---|---|---|---|---|---|---|
| 100 | Khasanah & Noordwijk (2018) | Site 03 | Oil palm plantation | P | Estate crop | Jambi | −1.53 | 102.37 | 96.13 | 29.65 | 74.98 | 23.13 | | 40 | 30 | 2349 | 0.14 |
| 101 | Khasanah & Noordwijk (2018) | Site 04 | Rubber plantation | P | Estate crop | Jambi | −1.53 | 102.37 | 75.17 | 3.55 | 58.63 | 2.77 | | 46 | 30 | 2349 | 0.19 |
| 102 | Marwanto & Agus (2014) | Site 01 | Oil palm plantation | P | Estate crop | Jambi | −1.72 | 103.88 | 46.1 | 30.02 | 35.96 | 23.42 | | 91 | 26.7 | 2349 | 0.21 |
| 103 | Marwanto et al., (2019) | Site 01 | Oil palm plantation | P | Estate crop | Riau | 0.73 | 101.72 | 44.66 | 25.63 | 34.83 | 19.99 | | 36 | 26.3 | 1830 | 0.25 |
| 104 | Saragi-Sasmito et al., (2019) | Site 01 | Secondary forest | F | Secondary swamp forest | Central Kalimantan | −2.92 | 114.42 | 52.11 | 4.04 | 40.74 | 3.3 | a | 110 | 27 | 1600 | 0.01 |
| 105 | Sundari et al., (2012) | Site 01 | Drained forest | D/B | Secondary swamp forest | Central Kalimantan | −2.53 | 114.5 | 44.92 | 14.08 | 35.04 | 10.98 | | 51 | 26.2 | 2005 | na |
| 106 | Sundari et al., (2012) | Site 03 | Undrained forest | F | Primary swamp forest | Central Kalimantan | −2.53 | 114.5 | 49.39 | 13.52 | 38.52 | 10.55 | | 9 | 26.2 | 2005 | na |
| 107 | Swails et al., (2019) | Site 01 | Oil palm plantation | P | Estate crop | Central Kalimantan | −2.78 | 111.8 | 55.9 | 13.58 | 43.6 | 10.59 | | 50 | 27.4 | 2058 | 0.34 |
| 108 | Swails et al., (2019) | Site 02 | Oil palm plantation | P | Estate crop | Central Kalimantan | −2.78 | 111.8 | 79.5 | 15.7 | 62.01 | 12.25 | | 50 | 27.4 | 2058 | 0.34 |
| 109 | Swails et al., (2019) | Site 03 | Oil palm plantation | P | Estate crop | Central Kalimantan | −2.78 | 111.8 | 49.1 | 19.94 | 38.3 | 15.55 | | 50 | 27.4 | 2058 | 0.34 |
| 110 | Swails et al., (2019) | Site 04 | Primary forest | F | Primary swamp forest | Central Kalimantan | −2.78 | 111.8 | 42 | 6.36 | 32.76 | 4.96 | | 23 | 27.4 | 2058 | 0.2 |
| 111 | Swails et al., (2019) | Site 05 | Primary forest | F | Primary swamp forest | Central Kalimantan | −2.78 | 111.8 | 39.4 | 11.88 | 30.73 | 9.27 | | 23 | 27.4 | 2058 | 0.2 |
| 112 | Swails et al., (2019) | Site 06 | Secondary forest | F | Secondary swamp forest | Central Kalimantan | −2.78 | 111.8 | 54.3 | 16.12 | 42.35 | 12.57 | | 23 | 27.4 | 2058 | 0.2 |
| 113 | Toma et al., (2011) | Site 01 | Crop- and grassland | C/S | Mixed dry agriculture | Central Kalimantan | −2.28 | 114.02 | 108.41 | 35.22 | 84.56 | 27.47 | | 75 | 25.9 | 2734 | 0.4 |
| 114 | Wakhid et al., (2017) | Site 01 | Rubber plantation | P | Estate crop | Central Kalimantan | −2.48 | 114.19 | 120.74 | 38.11 | 51.63 | 7.85 | a | 69 | 26.9 | 2506 | 0.23 |
| 115 | Watanabe et al., (2009) | Site 01 | Sago plantation | P | Estate crop | Riau | 0.85 | 102.77 | 13.81 | 6.76 | 10.77 | 5.27 | | 82 | 27.6 | 1700 | na |
| 116 | Watanabe et al., (2009) | Site 02 | Sago plantation | P | Estate crop | Riau | 0.85 | 102.77 | 13.81 | 8.03 | 10.77 | 6.26 | | 82 | 27.6 | 1700 | na |
| 117 | Watanabe et al., (2009) | Site 03 | Sago plantation | P | Estate crop | Riau | 0.85 | 102.77 | 15.74 | 8.03 | 12.28 | 6.26 | | 82 | 27.6 | 1700 | na |
| 118 | Watanabe et al., (2009) | Site 04 | Sago plantation | P | Estate crop | Riau | 0.85 | 102.77 | 17.02 | 5.46 | 13.28 | 4.26 | | 82 | 27.6 | 1700 | na |

Remarks: Latitude and longitude are in degrees (°), T-CO$_2$: total CO$_2$ emissions (Mg ha$^{-1}$ year$^{-1}$), H-CO$_2$: heterotrophic CO$_2$ emissions (Mg ha$^{-1}$ year$^{-1}$), SE = standard error, WTD: water table depth (cm), Temp.: air temperature (°C), Rainfall: annual rainfall (mm year$^{-1}$), BD: bulk density (g cm$^{-3}$), na: not available, a: actual data of heterotrophic emissions.

**Table A3.** Summary statistics of water table depth and bulk density for each land cover class as defined by Prananto et al. [61].

| Predictor Variable | Land Use | *n* | Mean | SD | Min. | Max. |
|---|---|---|---|---|---|---|
| Water table depth (*W*) | CS | 15 | 45.75 | 34.14 | −2 | 108 |
| | DB | 25 | 45.96 | 28.72 | 6 | 117.5 |
| | F | 22 | 32.51 | 29.37 | −5 | 110 |
| | P | 45 | 61.61 | 21.97 | 21.8 | 119 |
| Bulk density (*B*) | CS | 6 | 0.39 | 0.03 | 0.33 | 0.42 |
| | DB | 19 | 0.37 | 0.41 | 0.09 | 1.19 |
| | F | 15 | 0.29 | 0.36 | 0.01 | 1.17 |
| | P | 34 | 0.23 | 0.25 | 0.06 | 1.18 |

Remarks: *n* = number of observations, SD = standard deviation, Min. = minimum value, Max. = maximum value.

**Table A4.** Summary statistics of water table depth and bulk density for each land cover class as defined by the MOEF [62].

| Variable | Land Cover | *n* | Mean | SD | Min. | Max. |
|---|---|---|---|---|---|---|
| Water table depth (*W*. cm) | Bare ground (BG) | 3 | 70 | 3.61 | 67 | 74 |
| | Estate crop (EC) | 33 | 57.94 | 20.42 | 26 | 119 |
| | Mixed dry agriculture (MDA) | 5 | 65.16 | 27.15 | 21.8 | 93 |
| | Paddy field (PDF) | 3 | 5.4 | 7.5 | −2 | 13 |
| | Plantation forest (PF) | 10 | 78 | 18.8 | 36 | 108 |
| | Primary swamp forest (PSF) | 8 | 21.13 | 8.69 | 9 | 35 |
| | Pure dry agriculture (PDA) | 8 | 39.56 | 20.85 | 15 | 70 |
| | Secondary swamp forest (SSF) | 26 | 43.35 | 34.57 | −5 | 117.5 |
| | Wet shrub (WS) | 11 | 42.36 | 28.84 | 6 | 108 |
| Bulk density (*B*. g cm$^{-3}$) | Bare ground (BG) | 3 | 0.12 | 0 | 0.12 | 0.12 |
| | Estate crop (EC) | 22 | 0.3 | 0.29 | 0.09 | 1.18 |
| | Mixed dry agriculture (MDA) | 5 | 0.3 | 0.13 | 0.14 | 0.4 |
| | Plantation forest (PF) | 11 | 0.19 | 0.33 | 0.06 | 1.19 |
| | Primary swamp forest (PSF) | 9 | 0.38 | 0.45 | 0.11 | 1.17 |
| | Pure dry agriculture (PDA) | 2 | 0.4 | 0.03 | 0.38 | 0.42 |
| | Secondary swamp forest (SSF) | 13 | 0.39 | 0.43 | 0.01 | 1.14 |
| | Wet shrub (WS) | 9 | 0.18 | 0.07 | 0.13 | 0.33 |

Remarks: *n* = number of observations, SD = standard deviation, Min. = minimum value, Max. = maximum value.

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
