# Peer review of "Geographic Setting and Groundwater Table Control Carbon Emission from Indonesian Peatland: A Meta-Analysis"

_forests, doi:10.3390/f12070832_

Round 1

Reviewer 1 Report

forests-1215288

Manuscript title Geographic setting and groundwater table control carbon emission from Indonesian peatland: A Meta-Analysis “ is in line with the aim of the journal. Paper could be very interesting for the reader of Forest. I have some issues with these paper and those should be clarified before the paper would be accepted.

  • Line 16-17 NDC (Nationally Determined Contribution) should be written as NDC (Nationally Determined Contribution)
  • Line 21 …. Is also calculated…….was also calculated
  • Line 26, 27 and other places - Units of measurement are not as per the MDPI format
  • Line 30- Keywords should be arranged alphabetically
  • Line 36 UNFCCC – Abbreviations should be given in bracket
  • The introduction part is very long- Should be curtailed
  • The methodology should be described under various subheading
  • 1. Scope of this study – should be given in brief
  • Table 1 Column 2 – Author name and year should be followed by Reference No.
  • Table 1 – Give only recent data – 2015-2021
  • Table 2 – Column 1 – Should have full left margin justification
  • Results and Discussion should be given separately

Reviewer 2 Report

This study used a meta-analysis to provide an emission factor estimate for peat decomposition from the peatland literatures in Indonesia. The modeling work has been carefully explored, and estimates were generated. I like the detailed explanation of the methods and the honesty and integrity in data mining. However, I have a few concerns which for the authors to address:    

LINE/s

COMMENTS

16-17

‘Peat restoration is one of key of climate mitigation actions as stated on Indonesia’s NDC (Nationally Determined Contribution)’ considering revising, hard to understand

114-115

‘Land management classes in this study include managed, unmanaged, and natural’. How do you classify unmanaged from natural? Isn’t it that natural forests are also unmanaged sites?

120-135

In line 122, authors stated ‘Additional data were also extracted from the publications to provide predictor variables….’ in lines 131-133 authors also stated ‘In addition to total CO2 and heterotrophic emissions, we also collect environmental variables data including water table depth, air temperature, soil temperature, latitude longitude, rain fall, soil bulk 133 density, carbon and nitrogen contents’. How do these two collection process differ? The first one were extracted from literatures, the second one were observed variables or still a collection from publications? They seem to have similar variables being collected. Please enlighten the readers.

142-144

‘In 142 such case, the samples plot data were then analyzed to generate mean and SE of CO2 143 emissions for each site that represented a certain land use or land cover based on the 144 MoEF or IPCC Wetland (2014) classes’. By how?

164

Why use absolute latitude? How will it affect the true spatial location of the sites?

186-187

I am critical about only 25% in the publications provided heterotrophic data that may result to huge uncertainty in the results. Please justify.

Table 3

You inferred heterotrophic respiration to be 78% of the total respiration but this ratio was only obtained in 25% of the observation. I wonder if these 25% are even sparsely distributed all throughout Indonesia.  

209-211

‘These missing data could not be inferred 209 from the publications because the authors did not measure all of the variables used in this 210 study, namely water table depth and bulk density’. How did you gapfill these variables?

220-223

Have you tried comparing this value with other peatland sites other than in Indonesia?

238

‘CO2’ into ‘CO2’ please find more in the text

249

‘32.20-82.73’, please use long dash

248-249

‘The results 248 shows that the emission factor in land has range 32.20-82.73 Mg CO2 ha-1 y-1.’ If you try to look at your Total CO2,it is only 48.22 Mg CO2 221 ha-1 y-1 (95% CI: 42.36– 54.08 Mg CO2 ha-1 y-1). So, the upper range of heterotrophic CO2 exceeded that of Total CO2. Please explain. This may be due to the limited data on heterotrophic respiration and the gap-filling done.

249-250

Be careful in drawing conclusion, you have to caution the reader on your drawbacks, flaws and limitations on how you derived heterotrophic respiration.  

419-420

Must at least discuss why land use category has no effect on total and heterotrophic emissions

Round 2

Reviewer 1 Report

Author has improved the manuscript as per my suggestion. I recommend accepting the manuscript in the present form.